# Impact of MgtC on the Fitness of *Yersinia pseudotuberculosis*

**DOI:** 10.3390/pathogens12121428

**Published:** 2023-12-08

**Authors:** Peng Li, Hengtai Wang, Wei Sun, Jiabo Ding

**Affiliations:** 1Institute of Animal Science, Chinese Academy of Agricultural Sciences, Beijing 100193, China; 17861509833@163.com; 2Department of Immunology and Microbial Disease, Albany Medical College, Albany, NY 12208, USA; sunw@amc.edu

**Keywords:** *Yersinia pseudotuberculosis*, MgtC, intracellular survival, pathogenic

## Abstract

*Yersinia pseudotuberculosis* is an extracellular foodborne pathogen and usually causes self-limiting diarrhea in healthy humans. MgtC is known as a key subversion factor that contributes to intramacrophage adaptation and intracellular survival in certain important pathogens. Whether MgtC influences the fitness of *Y. pseudotuberculosis* is unclear. According to in silico analysis, MgtC in *Y. pseudotuberculosis* might share similar functions with other bacterial pathogens, such as *Salmonella*. Studies indicated that MgtC was clearly required for *Y. pseudotuberculosis* growth in vitro and bacterial survival in macrophages under Mg^2+^ starvation. Transcriptome analysis by RNA-seq indicated that 127 differentially expressed genes (DEGs) (fold change > 2 and *p* < 0.001) were discovered between wild-type PB1+ and *mgtC* mutant inside macrophages. However, a lack of MgtC only moderately, albeit significantly, reduced the virulence of *Y. pseudotuberculosis* in mice. Overall, this study provides additional insights for the role of MgtC in *Y. pseudotuberculosis*.

## 1. Introduction

*Yersinia pseudotuberculosis*, a Gram-negative foodborne pathogen, has wide host spectrum and can circulate between human and animals [1,2,3]. *Y. pseudotuberculosis* causes yersiniosis in humans, ranging from mild diarrhea, enterocolitis, and lymphatic adenitis to persistent local inflammation [4]. *Y. pseudotuberculosis* is generally considered to be an extracellular pathogen [5], but a growing body of evidence has shown that *Y. pseudotuberculosis* also survives and replicates inside macrophages, which may contribute to its initial infection phase [6,7,8].

Bacterial pathogens have developed strategies to evade the innate immune system and counteract the microbicidal action of macrophages [9]. Intracellular bacterial pathogens use a broad range of molecular determinants to manipulate host cell processes and adapt to the intracellular environment [10]. MgtC, an integral membrane protein, is a key subversion factor that contributes to intramacrophage adaptation and intracellular survival of a variety of bacterial pathogens [11]. *Salmonella enterica*, *Mycobacterium tuberculosis*, and other intracellular bacterial pathogens rely on the MgtC protein to survive within acidic macrophage phagosomes and cause a lethal infection in mice [12,13,14,15]. Importantly, MgtC has been shown to promote intramacrophage survival of extracellular pathogens *Pseudomonas aeruginosa* and *Yersinia pestis* during the intracellular stage [13,16]. For the intracellular pathogen *Salmonella*, MgtC inhibits the activity of the F_1_F_o_ ATP synthase by direct interaction, hindering ATP-driven proton translocation and NADH-driven ATP synthesis in inverted vesicles [17]. Moreover, MgtC promotes *Salmonella* virulence by limiting cellulose production during infection [18] and phosphate uptake of *Salmonella* inside macrophages [19].

In addition to its role in promoting intramacrophage survival, MgtC promotes the growth of bacteria in acidic environments and under magnesium (Mg^2+^) starvation [20]. Mg^2+^ is critical for bacterial survival and replication in vivo [21]. However, MgtC is not necessary for Mg^2+^ transport [20,22]. In *Salmonella*, growth in low-Mg^2+^ media also promotes *mgtC* expression, even when *Salmonella* experiences a neutral pH [23]. Increased MgtC allows *Salmonella* to promote the transcription of *Pho* genes in response to the PhoP/PhoQ-activating signals, such as low Mg^2+^, low phosphate, or under acidic conditions [19]. However, it is unclear whether the MgtC of *Y. pseudotuberculosis* has similar functions as other pathogens in vitro and in vivo. Our results indicated that *Y. pseudotuberculosis* MgtC promoted bacterial growth under Mg^2+^ starvation in vitro and was essential for bacterial intramacrophage survival and replication. Disruption of MgtC diminished the fitness of *Y. pseudotuberculosis* in mice to a moderate extent.

## 2. Material and Method

### 2.1. Strains, Plasmids, Macrophages and Culture Conditions

The bacterial strains and plasmids used in this study were described in Table 1. *Escherichia coli* DH5α as a routine cloning strain and *E. coli* χ7213 as a suicide plasmid donor strain were cultured at 37 °C in Luria Bertani (LB) broth or on LB agar plates supplemented with 50 μg/mL diaminopimelic acid (DAP) or 25 μg/mL chloramphenicol (Cm) as necessary. *Y. pseudotuberculosis* and its derivatives were grown on LB agar or LB broth at 28 °C. LB plates containing 5% sucrose were used for *sacB* gene-based counter-selection in allelic exchange experiments for construction of *Y. pseudotuberculosis* mutants. When appropriate, 100 μg/mL ampicillin and/or 25 μg/mL chloramphenicol were added. Murine macrophage RAW264.7 (ATCC, Manassas, VA, USA) was cultured at 37 °C with 5% CO_2_, in Dulbecco’s modified Eagle’s medium (DMEM) supplemented with 10% heat-inactivated fetal bovine serum (Gibco, Grand Island, NY, USA).

### 2.2. Plasmid Construction

All primers used in this study were listed in Appendix A. Suicide plasmids were constructed using an overlap PCR as previously reported [27]. Briefly, the upstream and downstream fragments of each target gene amplified by PCR and purified from agarose gels as templates were spliced by an overlap PCR. The overlap PCR product containing the flanking sequences of each target gene was purified by gel extraction Kit (Zymo Research, CA, USA) and cloned into *Kpn*I and *Xma*I sites of the pRE112 plasmid to generate a suicide plasmid for deleting each corresponding gene. To complement the function of each deleted gene, the target gene containing its native promoter was amplified by PCR using the corresponding primer set and then cloned into a low-copy plasmid pYA4454 (pSC101 *ori*) to generate the corresponding complementary plasmid.

### 2.3. Strain Construction

The procedures for constructing *Y. pseudotuberculosis* mutant were described in previous studies [5,25]. Briefly, the suicide plasmid pSMV33 (Δ*mgtC*) was conjugationally transferred from *E. coli* χ7213 [28] to WT PB1+ and PB1+ pYV-strains, respectively. Single-crossover insertion strains were isolated on LB agar plates containing Cm. Loss of the suicide vector after the second recombination between homologous regions (i.e., allelic exchange) was selected by using the *sacB*-based sucrose sensitivity counter-selection system. The colonies were screened for Cm^s^ (chloramphenicol sensitive) and verified by PCR using primers MgtC-1/MgtC-4. Also, the same method was followed to delete *YPTS_2313* and/or *YPTS_3071* from *Y. pseudotuberculosis*.

### 2.4. Determination of Bacterial Growth Curve in High or Low Mg^2+^ Minimal Medium

*Y. pseudotuberculosis* and its derived mutants were grown on 2 mL LB broth at 28 °C to optical density 600 nm (OD_600_) ranging from 0.6 to 0.8, and then collected by centrifugation at 8000× *g* for 5 min. M9 minimal salts medium supplemented with 0.4% (*v*/*v*) glucose as a carbon source and 0.1% (*v*/*v*) casamino acid was used for analyzing bacterial responses to Mg^2+^. The pellet of each strain was washed twice with M9 minimal medium and then resuspended in 2 mL of the same medium supplemented with 10 µM Mg^2+^ or 10 mM Mg^2+^. Each strain was incubated aerobically at 28 °C in an orbital shaker (200 rpm) for 24 h, and bacterial growth curves were recorded by measurement of OD_600_ over the course of incubation period.

### 2.5. RNA Extraction and Real-Time PCR

Total RNA was extracted from bacteria using the Quick-RNA™ Miniprep Plus Kit (Zymo Research) according to the manufacturer’s protocol. Genomic DNA contamination was removed through treatment with a Turbo DNA-free kit (Ambion, Carlsbad, CA, USA). The RNA quantity and quality were evaluated by calculation of RNA concentration and OD_260_/OD_280_ ratio (1.8–2.0) using the NanoDrop ND-1000 spectrophotometer (NanoDrop Technologies, Wilmington, DE, USA), and the RNA integrity was assessed by calculation of RNA integrity number (RIN) using standard denaturing agarose gel electrophoresis. RNA (1 μg) was then reversely transcribed into cDNA using the One*Taq*^@^ RT-PCR Kit (New England BioLabs, Beverly, MA, USA) according to the manufacturer’s instructions. Luna^®^ Universal qPCR Master Mix (New England BioLabs, Beverly, MA, USA) was used for qRT-PCR, according to the manufacturer’s instruction: 10 μL Luna Universal qPCR Master Mix, 1 μg cDNA, 0.5 μL forward or backward primer (10 μM), and 8 μL nuclease-free water were added. The reaction system was incubated at 95 °C for 1 min, and then subjected to 40 cycles at 95 °C for 15 s, followed by 60 °C for 30 s, and then followed a melt curve using a Mastercycler^®^ ep Realplex system (Eppendorf, Hamburg, Germany). All samples were analyzed in triplicate and relative transcription levels of each gene were determined by the 2^−ΔΔCt^ method, using *rpoD* as an internal control for data normalization.

### 2.6. Bacterial Intracellular Survival Assay

Bacterial intracellular survival was tested using RAW 264.7 cells. Infection assays were performed as previously described [27]. Briefly, monolayers of cells (5 × 10^5^ cells) were cultured in 24-well plates and infected *Y. pseudotuberculosis* and its derivatives at a multiplicity of infection (MOI) of 20 colony forming units (CFUs) per cell. To synchronize the infection, infected plates were centrifuged at 400× *g* for 5 min followed by 37 °C with 5% CO_2_ for 1 h. Cells were washed twice with PBS to remove nonadherent bacteria, and then incubated with DMEM containing 100 μg/mL gentamicin for 1 h to kill extracellular bacteria. The time point after killing extracellular bacteria for 1 h was set as 0 h post-infection (p.i.). To maintain survival of infected cells, monolayers were incubated with DMEM containing 20 μg/mL gentamicin and 2% FBS after being washed thrice with PBS followed by 37 °C with 5% CO_2_. At 0 h, 4 h, 8 h, and 24 h p.i., infected cells were washed and incubated with 1 mL of 0.2% (*v*/*v*) Triton X-100 water solution for 10 min at 37 °C. Then, cell suspension was serially diluted 10-fold with PBS and spread onto LB agar plates to enumerate viable bacteria (CFUs).

### 2.7. Lactate Dehydrogenase (LDH) Activity Assay

RAW264.7 cells were infected with WT PB1+ or Δ*mgtC* mutant as described above. Cell culture supernatant was harvested at 24 h p.i. LDH activity was measured in the cell culture medium using the LDH Activity Assay Kit (Sigma-Aldrich, St. Louis, MO, USA; Catalog Number: MAK066) according to the manufacturer’s instructions.

### 2.8. RNA-Seq

Total RNA was extracted from WT PB1+ or Δ*mgtC* mutant infecting RAW264.7 cells, and then RNA-seq was performed. Briefly, RAW264.7 cells were infected with WT PB1+ and Δ*mgtC* mutant, respectively, as described above, and then at 2 h p.i., infected cells were washed and lysed with 1 mL of 0.2% (*v*/*v*) Triton X-100 water solution for 10 min at 37 °C. The lysis solution was centrifuged at 8000× *g* for 5 min to precipitate bacteria. Total RNA was extracted from bacteria as described above. Ribosomal RNA was removed using the RiboZero kit (Illumina, San Diego, CA, USA). RNA-seq libraries were prepared using the ScriptSeq 2.0 Kit (Illumina, San Diego, CA, USA). Sequencing was performed using an Illumina Next-Seq instrument (GENEWIZ, Suzhou, China). Differential RNA expression analysis was performed using Rockhopper (version 2.03) with default parameters [29]. Differences in RNA levels were considered to indicate regulation for genes with false discovery rate (*q*) values of ≤0.01 and fold-change values ≥2.

### 2.9. Animal Infection

Animal care and experimental protocols were in accordance with the NIH “Guide for the Care and Use of the laboratory Animals” and were approved by the Institutional Animal Care and Use Committee at Albany Medical College (IACUC protocol# 20-01001).

Six-week-old, female Balb/c mice were purchased from Charles River Laboratories (Wilmington, MA, USA). Mice were acclimated for one week before experiments. Overnight-grown cultures of WT PB1+ and its derived mutants were re-inoculated in fresh LB broth, respectively. Bacterial cultures were grown at 28 °C with 180 rpm constant agitation until exponential phase (OD_600_ 0.8–0.9). Bacterial cells were pelleted at 4000× *g* for 12 min and resuspended in sterile phosphate-buffered saline (PBS), pH 7.4, with an appropriate volume. Groups of mice (10 mice each group) were deprived of food and water for 6 h and then gavaged 0.2 mL bacterial suspension to each mouse with a single dose of 5.0 × 10^8^ colony forming units (CFU) by using 20-gauge feeding needle. Mortality and morbidity of infected mice were observed daily for the next 15 days.

The kinetics of bacterial burden in different vital organs including intestine, spleen, liver, and lung were assessed at 2, 4, and 6 days p.i. Three mice from each group were humanly euthanized and perfused with 5 mL sterile PBS before collecting lung, liver, spleen, and Peyer’s patch, aseptically. Tissue samples were homogenized by bullet blender (Bullet Blender Blue; Averill Park, NY, USA) and serially diluted homogenates were spread on *Yersinia* selective agar plates [Cefsulodin, Irgasan, Novobiocin (CIN) agar] in duplicates. After 48 h incubation at 28 °C, CFU were counted and calculated according to the initial weight (mg) and/or volume (mL) of the organ [5].

### 2.10. Statistical Analysis

Data were analyzed using GraphPad PRISM 8.0 software. Statistical analyses of data were evaluated by two-way ANOVA using Tukey’s post hoc tests. The log-rank test was used for analysis of the survival curves. All results are presented as means ± standard deviations (SD) for at least three independent experiments, and data represented significance at * *p* < 0.05, ** *p* < 0.01, *** *p* < 0.001, **** *p* < 0.0001.

## 3. Results

### 3.1. Phylogenetic Analysis of MgtC-like Proteins in Different Bacteria

In silico analysis showed that YPTS_2490 in *Y. pseudotuberculosis* belonged to the MgtC/SapB family protein and the same phylogenetic subgroup as MgtC of *Salmonella typhimurium* (Figure 1). A further phylogenetic analysis using the neighbor-joining tree (NJ) method of molecular evolutionary genetics analysis (MEGA) Software revealed that MgtC in *Y. pseudotuberculosis* had a close relationship with MgtC of other facultative intracellular pathogens, namely *Pseudomonas aeruginosa*, *Burkholderia cenocepacia*, *Mycobacterium tuberculosis,* and *E. coli* (Figure 1). To identify the molecular characteristics of MgtC in *Y. pseudotuberculosis* and other intracellular pathogens, the deduced amino acid sequences of each protein were analyzed and aligned using DNASTAR software (v7.1). The amino acid sequence of *Y. pseudotuberculosis* MgtC is 63% identical to the MgtC of *S. typhimurium*, 45% identical to the MgtC of *B. cenocepacia*, 54% identical to the MgtC of *P. aeruginosa*, and 39% identical to the MgtC of *M. tuberculosis*, respectively. Additionally, there is no difference for the MgtC/SapB family protein sequence in the different strains of *Y. pseudotuberculosis*, such as *Y. pseudotuberculosis* serotype O:1b (strain PB1/+), *Y. pseudotuberculosis* serotype O:1b (strain IP 31758), *Y. pseudotuberculosis* serotype I (strain IP32953), and *Y. pseudotuberculosis* serotype O:3 (strain YPIII). These results suggested that YPTS_2490 was considered as MgtC in *Y. pseudotuberculosis* and might share similar functions as it in other bacterial pathogens.

### 3.2. MgtC Is Required for Y. pseudotuberculosis Growth under Mg^2+^ Starvation and mtgC Transcription Is Strongly Induced under Mg^2+^ Starvation and Intracellular Conditions

A previous study showed that MgtC is essential for *Salmonella* growth under Mg^2+^ starvation [20]. To determine whether MgtC of *Y. pseudotuberculosis* is necessary for the growth of bacteria under Mg^2+^ starvation, we constructed an in-frame deletion of *mgtC* in *Y. pseudotuberculosis* to avoid any polar effect on downstream genes and complemented the *mgtC* mutation with a very low-copy plasmid, pSMV58 constitutively expressing *mgtC* (Table 1). Then, the growth of WT PB1+ and its derived Δ*mgtC* mutants were compared under low and high Mg^2+^ conditions. Under in vitro growth conditions, we found that the Δ*mgtC* mutant was deficient in growth under the low-Mg^2+^ media and that the growth defect in the Δ*mgtC* mutant was completely restored by MgtC complementation (Figure 2A). However, the Δ*mgtC* mutant did not display a growth defect in the high-Mg^2+^ media compared with the growth of WT PB1+ and the MgtC complementary strain (Figure 2B). These results indicated that MgtC was required for optimal growth of *Y. pseudotuberculosis* under Mg^2+^ starvation in vitro. 

To determine whether the *Y. pseudotuberculosis mgtC* is regulated by Mg^2+^ starvation, we compared *mgtC* expression of WT PB1+ strain under low- and high-Mg^2+^ conditions by qRT-PCR. The *mgtC* expression in WT PB1+ strain under the low-Mg^2+^ condition was dramatically increased compared to that under the high-Mg^2+^ condition, suggesting that *Y. pseudotuberculosis mgtC* expression is strongly induced under the low-Mg^2+^ condition (Figure 2C).

Extracellular *Y. pseudotuberculosis* can survive and replicate inside macrophages [6,8]. Macrophages contain low cytosolic Mg^2+^ concentration and acidic environments [30]. To determine whether the *Y. pseudotuberculosis mgtC* is regulated under intracellular condition, we compared the *mgtC* expression of WT PB1+ strain under extracellular and intracellular conditions by qRT-PCR. The *mgtC* expression of WT PB1+ strain within macrophages was significantly increased in comparison to that under extracellular condition, suggesting that the *Y. pseudotuberculosis mgtC* expression is strongly induced under intracellular condition (Figure 2D). Taken together, MgtC promoted growth of *Y. pseudotuberculosis* under Mg^2+^ starvation, and the *mgtC* expression was strongly induced within macrophages and under low-Mg^2+^ conditions.

### 3.3. MgtC Plays a Critical Role for Y. pseudotuberculosis Replication in Macrophages

The MgtC of an intracellular pathogen, *Salmonella enterica*, is required for its intramacrophage survival [12]. To determine whether MgtC is involved in the intracellular survival of extracellular pathogen *Y. pseudotuberculosis*, we compared the survival of WT PB1+, a Δ*mgtC* mutant, and a *mgtC* complementary strain inside RAW264.7 macrophages. RAW264.7 macrophages were infected with each strain, and then intracellular bacteria were recovered from infected macrophages at different times post-infection (p.i.). Results showed that three strains had similar bacterial titers at 0 h p.i., suggesting that the presence or absence of MgtC does not affect the invasion of *Y. pseudotuberculosis*. The number of Δ*mgtC* mutant inside RAW264.7 cells dramatically decreased starting at 4 h p.i. in comparison to with WT PB1+ and Δ*mgtC* mutant complemented with *mgtC* (Figure 3A), suggesting that MgtC facilitates the intracellular replication of the primarily extracellular *Y. pseudotuberculosis* when it is engulfed by macrophages, as well as suggesting that the decreased intracellular survival of the Δ*mgtC* mutant is caused by a lack of *mgtC* rather than an eventual polar effect caused by the mutation. The number of intracellular WT PB1+ strains increased over the course of the infection from 0 h to 8 h p.i., but significantly decreased at 24 h p.i. (Figure 3A), suggesting that *Y. pseudotuberculosis* as a primarily extracellular pathogen can replicate in the macrophages at an early stage, but cannot sustain for 24 h p.i.

*Y. pseudotuberculosis* contains virulence plasmid pYV that encodes the type three secretion system (T3SS) [31], which is responsible for injecting a number of *Yersinia* outer proteins (Yops) into host cells. Yops can inhibit bacterial phagocytosis, the respiratory burst, and the host innate immune response, and trigger apoptosis [32,33]. Intriguingly, the *Y. pseudotuberculosis* strains’ lack of virulence plasmid pYV did not change their survival patterns in macrophages (Figure 3B), indicating that MgtC facilitates intracellular replication of extracellular pathogen *Y. pseudotuberculosis* independent of the presence of pYV.

To test the cytotoxicity of the Δ*mgtC* mutant on macrophages, the release of LDH was determined for PB1±, Δ*mgtC*−, and Δ*mgtC*(*C*−*mgtC*) −infected macrophages. There were no significant differences in levels of LDH released from the PB1±, Δ*mgtC*−, and Δ*mgtC*(*C*−*mgtC*) −infected macrophages at 24 h p.i. (Appendix A), indicating that the *mgtC* deletion has no effect on cytotoxic action of *Y. pseudotuberculosis* towards macrophages.

### 3.4. Genes Analysis of mgtC Mutant within Macrophages by RNA-Seq

Given significant survival differences in macrophages, we attempted to explore how MgtC regulated gene expression profiles of *Y. pseudotuberculosis* within macrophages. The total RNA was extracted from PB1+ or the Δ*mgtC* mutant strain released from infected RAW264.7 cells, and then RNA-seq was performed to characterize differentially expressed genes (DEGs). We found 127 DEGs (fold-change > 2 and *p* < 0.001) under the intracellular condition. Among them, 98 genes were significantly upregulated, and 29 genes were significantly downregulated, while the *mgtC* gene was not detected in the Δ*mgtC* mutant strain (Appendix A). The hierarchical clustering heatmap (Figure 4A), the volcano plot (Figure 4B), principal component analysis (Figure 4C). and sample distance (Figure 4D) all displayed clear DEGs between WT PB1+ and Δ*mgtC*.

Among 127 DEGs, the top 10 most significant DEGs are shown in Table 2. To confirm the authenticity of RNA-seq, transcriptional levels of top 10 most significant DEGs were measured in WT PB1+ and Δ*mgtC* mutant isolated from the infected RAW264.7 cells by qRT-PCR. The results of qRT-PCR for those DEGs were consistent with the RNA-seq results (Figure 4E). Among the 10 most significant DEGs, *YPTS_2313* and *YPTS_3071*, encoding acid shock proteins, were significantly downregulated in the Δ*mgtC* mutant isolated from infected RAW264.7 cells compared with WT PB1+. Previous studies showed that acid shock proteins were required for adaptive acid tolerance response in *Salmonella* and MgtC promoted the growth of bacteria in acidic environments [20,34]. We speculated that MgtC disruption in *Y. pseudotuberculosis* might downregulate *YPTS_2313* and *YPTS_3071*, resulting in decreased intracellular survival. To confirm the roles of *YPTS_2313* and *YPTS_3071* for *Y. pseudotuberculosis* intracellular survival, we constructed the following *Y. pseudotuberculosis* mutant strains: Δ*YPTS_2313*, Δ*YPTS_3071* and Δ*YPTS_2313*Δ*YPTS_3071*, respectively, and compared their survival rates inside RAW264.7 macrophages. However, results showed that the Δ*YPTS_2313*, Δ*YPTS_3071*, and Δ*YPTS_2313*Δ*YPTS_3071* mutants had similar survival rates to WT PB1+ (Appendix A), suggesting that Δ*YPTS_2313* and/or Δ*YPTS_3071* are not essential to the intracellular survival of *Y. pseudotuberculosis*.

### 3.5. Pathogenicity of the ΔmgtC Mutant In Vivo

For intracellular pathogens, MgtC is a key virulence factor and promotes their pathogenicity in vivo [17,35,36]. Therefore, to evaluate the role of MgtC in *Y. pseudotuberculosis* pathogenicity, Balb/C mice (10 mice per group) were infected with 5 × 10^8^ CFUs of WT PB1+ and the Δ*mgtC* mutant by oral gavage, respectively. All mice infected with WT PB1+ succumbed within 15 days post-administration, while mice infected with the Δ*mgtC* mutant had 40% survival (Figure 5A), suggesting that the *mgtC* mutation results in moderate attenuation of *Y. pseudotuberculosis*. The virulence of the Δ*mgtC* mutant was completely restored by the *mgtC* complementary plasmid (Figure 5A).

Following this, we compared the burdens of the Δ*mgtC* mutant in the Peyer’s patches, spleens, livers, and lungs at day 2, 4, and 6 p.i. with burdens of WT PB1+. Bacterial titers of both WT PB1+ and the Δ*mgtC* mutant in Peyer’s patches, spleens, livers, and lungs dramatically increased at day 4 p.i. compared with day 2 p.i. and was then maintained at day 6 p.i. in Peyer’s patches (Figure 5B–E). Titers of both WT PB1+ and the Δ*mgtC* mutant slightly increased at day 6 p.i. in comparison to day 4 p.i. Bacterial titers in the spleens, livers, and lungs of mice infected with WT PB1+ or the Δ*mgtC* mutant showed no significant differences, but titers of the Δ*mgtC* mutant in a few mice were obviously lower than titers of WT PB1+ (Figure 5B–E). Taken together, the Δ*mgtC* mutant caused a slight attenuation in mice, and MgtC had marginal effects on the dissemination of *Y. pseudotuberculosis* in mice.

## 4. Discussion

Intracellular pathogens live inside host cells and require a myriad of factors to adapt to the harsh environments in the host cells. MgtC, first described in *S. typhimurium*, is a key player in bacterial intramacrophage survival, and is important for the virulence of diverse intracellular pathogens [11,37]. Moreover, the role of MgtC for macrophage survival is highly conserved in unrelated intracellular pathogens, such as *S. typhimurium*, *B. cenocepacia*, and *M. tuberculosis* [14,15,20,23]. A study showed that genes encoding MgtC-like proteins were found in a limited number of eubacterial genomes, and phylogenetic analysis suggested that *mgtC* were acquired by horizontal gene transfer repeatedly throughout bacterial evolution [38]. Based on sequence analysis, MgtC of *Y. pseudotuberculosis* had a close relationship to those of intracellular pathogens, such as *S. typhimurium* (Figure 1). As an extracellular pathogen, if *Y. pseudotuberculosis* were to acquire the *mgtC* gene via horizontal transfer from *S. typhimurium*, it might adapt to different environmental cues in mammalian hosts.

Mg^2+^ is involved in several cellular processes, such as the stabilization of membranes and ribosomes, as well as nucleic acid neutralization [39], while several intracellular pathogens require MgtC to grow in Mg^2+^-deprived environments [12,14,15,23]. In a similar fashion, the extracellular pathogen *Y. pseudotuberculosis* required MgtC to grow in Mg^2+^−deprived environments, and *mgtC* expression of *Y. pseudotuberculosis* was strongly induced under Mg^2+^ starvation (Figure 2A,C), indicating that *Y. pseudotuberculosis* MgtC is also involved in adaptation to low Mg^2+^ environments. Macrophages contain low cytosolic Mg^2+^ concentrations and acidic environments [30]. MgtC promotes the survival of several pathogens inside the macrophages by supporting bacterial adaptation to the intracellular environment [12,14,15,23]. Consistent with previous studies, *Y. pseudotuberculosis mgtC* expression was highly induced inside macrophages (Figure 2D) and is critical to survival of *Y. pseudotuberculosis* in macrophages (Figure 3A), suggesting that MgtC is required for the adaption of *Y. pseudotuberculosis* to the conditions encountered in macrophages.

There are conflicting reports about the ability of *Y. pseudotuberculosis* to replicate in macrophages. For example, YPIII, a serogroup O3 strain of *Y. pseudotuberculosis*, is unable to survive in mouse J774A.1 macrophages [40]. However, a serogroup O4b *Y. pseudotuberculosis* strain [41] and a serogroup O1 isolate, IP2790 [8], replicated in mouse peritoneal macrophages. Intriguingly, *Y. pseudotuberculosis* PB1+, a serogroup O1:b strain used in this study, presented different survival profiles in murine RAW 264.7 macrophages (Figure 3A). The number of viable intracellular WT PB1+ strains increased over the course of infection from 0 h to 8 h p.i., but significantly decreased at 24 h p.i. (Figure 3A). The discrepancy might be due to the different macrophages used in those studies. In addition, the pYV-cured PB1+ strain showed similar patterns of bacterial survival in macrophages when compared to WT PB1+ (Figure 3B), suggesting that the survival of the PB1+ strain in macrophages is not dependent on the virulence plasmid pYV. Our observations were consistent with the survival of intracellular *Y. pseudotuberculosis* IP2790 [8], but contrary to a previous report that the presence of a functional T3SS decreases the survival of intracellular *Y. pseudotuberculosis* YPIII [42]. Previous study mentioned that the ability of *Y. pseudotuberculosis* to replicate in macrophages might be a serogroup-specific property [8]. This explanation is also applicable to our observations. Further studies are needed to uncover unknown mechanisms.

Our study found that cellular survival of *Y. pseudotuberculosis* in macrophages was dependent on MgtC, but independent of the virulence plasmid pYV (Figure 3). The number of Δ*mgtC* mutants significantly decreased at 4 h p.i. and maintained a lower number of bacteria until 24 h p.i. (Figure 3). Interestingly, the Δ*mgtC* mutant with significantly lower numbers in macrophages caused similar levels of cytotoxic action towards macrophages as the WT PB1+ strain (Appendix A). The possible explanation is that some Δ*mgtC* mutant or PB1+ may be released to extracellular media during bacterial intracellular replication in macrophages, and extracellular replicated Δ*mgtC* mutant or PB1+ strains translocate Yops to macrophages, causing cell apoptosis and death [43]. The exact reasons will be pursued in a future study.

RNA-Seq and qRT-PCR analysis found that *YPTS_2313* and *YPTS_3071*, encoding acid shock proteins, were significantly downregulated in the *mgtC* mutant (Figure 4). The acid tolerance response of *S. typhimurium* involves transient synthesis of key acid shock proteins which are required for the adaptive acid tolerance response [34,44]. A low pH-inducible, PhoPQ-dependent acid tolerance response protects *S. typhimurium* against inorganic acid stress [45]. In *Salmonella*, the PhoP/PhoQ-activating signals, such as low Mg^2+^, low phosphate, or acidic conditions, are associated with MgtC [19]. However, our results showed that disruption acid shock protein (YPTS_2313 and/or YPTS_3071) did not influence on survival of *Y. pseudotuberculosis* in microphages or impact bacterial virulence in mice (Appendix A). One possible explanation is that *Y. pseudotuberculosis* might prefer to use the extracellular life cycle during infection, rather the intracellular life cycle. The Δ*mgtC Y. pseudotuberculosis* downregulated *YPTS_2313*, *YPTS_3071*, and other gene expressions (Table 2 and Appendix A), so moderate attenuation of the Δ*mgtC* mutant in mice might be due to an additive effect of those downregulated genes. In conclusion, MgtC is required for the in vitro growth and in vivo survival of *Y. pseudotuberculosis* under Mg^2+^ starvation, and disruption of MgtC diminished the fitness of *Y. pseudotuberculosis* in hosts to a moderate extent. This study provides an additional insight into MgtC function in extracellular pathogens.

## Figures and Tables

**Figure 1 pathogens-12-01428-f001:**
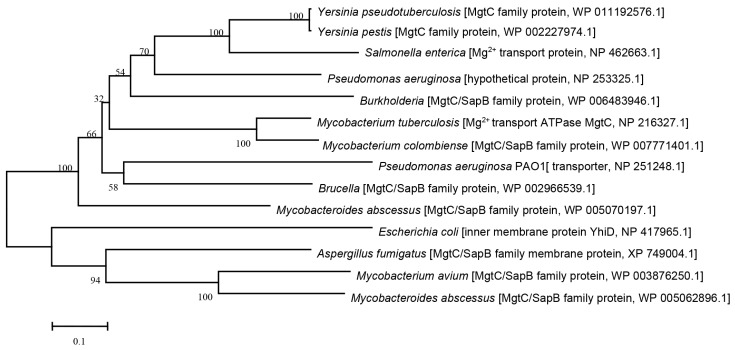
Phylogenetic analysis of *Y. pseudotuberculosis* MgtC and the reference strains using the NJ method of the MEGA software. MgtC from following bacteria: *Brucella melitensis*, *Burkholderia cenocepacia*, *Escherichia coli*, *Aspergillus fumigatus*, *Pseudomonas aeruginosa*, *Mycobacterium avium*, *M. abscessus*, *M. tuberculosis*, *Salmonella Typhimurium*, *Yersinia pestis*, and *Y. pseudotuberculosis* were included in this analysis.

**Figure 2 pathogens-12-01428-f002:**
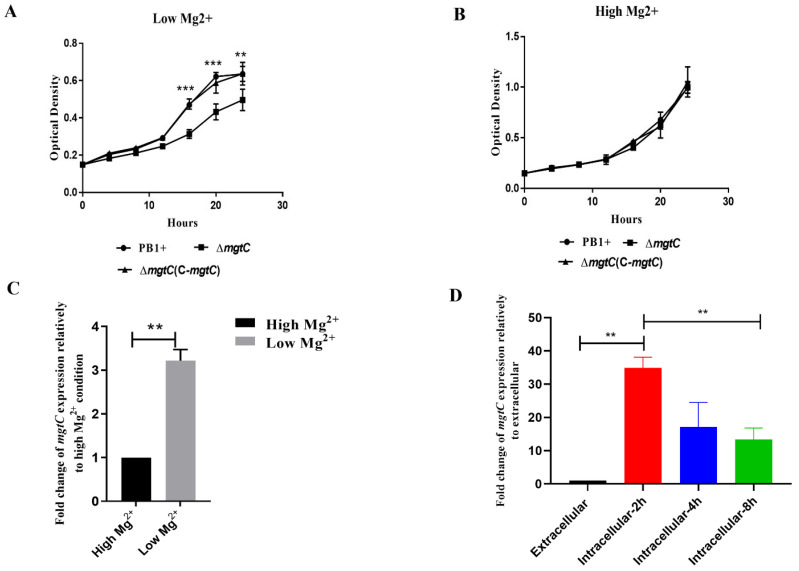
MgtC is required for *Y. pseudotuberculosis* growth under Mg^2+^ starvation and the *mtgC* transcription is strongly induced under Mg^2+^ starvation and intracellular conditions. (**A**) Growth curves of WT *Y. pseudotuberculosis* PB1+ (PB1+), Δ*mgtC* mutant (Δ*mgtC*), and Δ*mgtC* mutant complemented with *mgtC* [Δ*mgtC* (C-*mgtC*)] in low-Mg^2+^ medium. (**B**) Growth curves of PB1+, Δ*mgtC*, and Δ*mgtC*(C-*mgtC*) in high-Mg^2+^ medium. (**C**) qPCR analysis for the *mgtC* gene expression of WT PB1+ under a low-Mg^2+^ condition and a high-Mg^2+^ condition was detected. (**D**) qPCR analysis for the *mgtC* gene expression of WT PB1+ under extracellular media and intracellular RAW264.7 macrophages. Mean value bars ± SD. Statistical significance among groups were analyzed by a two-tailed *t* test. ** *p* ≤ 0.01; *** *p* ≤ 0.001.

**Figure 3 pathogens-12-01428-f003:**
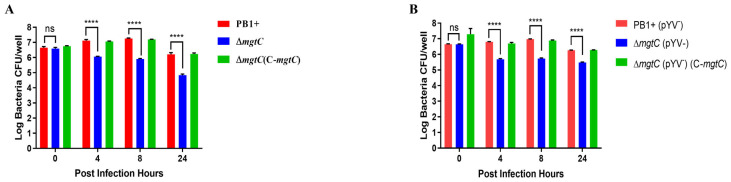
MgtC is required for *Y. pseudotuberculosis* replication in macrophages. (**A**) Intracellular survival within RAW 264.7 macrophages. RAW264.7 cells were infected with WT PB1+, Δ*mgtC*, and Δ*mgtC*(C−*mgtC*) at a MOI of 20. The infected cells were incubated with 0.2% (*v*/*v*) Triton X-100 at 0 h, 4 h, 8 h, and 24 h p.i. Then, intracellular CFUs of bacteria were enumerated. (**B**) Intracellular survival within RAW 264.7 macrophages. RAW264.7 cells were infected with each strain cured virulence plasmid pYV: WT PB1+(pYV^−^), Δ*mgtC*(pYV^−^), and Δ*mgtC*(C−*mgtC*)(pYV^−^) at a MOI of 20. The infected cells were incubated with 0.2% (*v*/*v*) Triton X-100 at 0 h, 4 h, 8 h, and 24 h p.i. Then, intracellular CFUs of bacteria were enumerated. Mean value bars ± SD. Statistical significance among groups were analyzed by two-way ANOVA multivariant applying Tukey’s post hoc test. ns, no significant, **** *p* ≤ 0.0001.

**Figure 4 pathogens-12-01428-f004:**
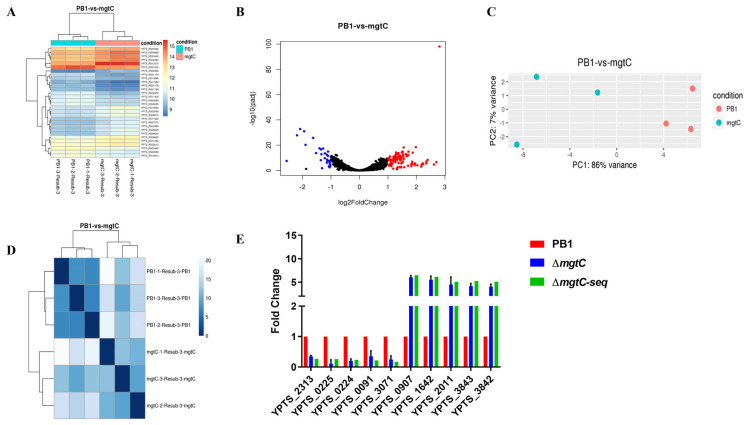
Transcriptome analyses of WT PB1+ and the Δ*mgtC* mutant within macrophages by RNA−seq. RAW264.7 cells were infected with WT PB1+ and Δ*mgtC*, respectively, at a MOI of 20. The infected cells were incubated with 0.2% (*v*/*v*) Triton X-100 at 2 h p.i.. The solution was centrifuged to precipitate bacteria. Total RNA was extracted from bacteria and the expression of DEGs were performed by RNA−seq. (**A**) Heatmap and hierarchical clusters analysis of the differentially expressed genes (DEGs) between WT PB1+ and Δ*mgtC* in macrophages (3 replicates each). Each small box indicates the expression status of a certain gene in one sample, with red for upregulated and blue for downregulated expression. (**B**) Volcano plot highlighting differentially expressed genes between unexposed and exposed bacteria. The genes are colored if they pass the thresholds for −log10 *p* value (*p* value = 0.05) and log−fold change |FC| ≥ 1.2, with red if they are upregulated and blue if they are downregulated. (**C**) Principal component analysis was applied to transcriptomic profiles of the WT PB1+ (pink circles) and Δ*mgtC* mutant (cyan-blue circles). Each dot represents an independent biological replicate. (**D**) Heat map showing hierarchical clustering of the Euclidean sample−to−sample distance between transcriptomic profiles of WT PB1+ and the Δ*mgtC* mutant. (**E**) Validation of RNA−seq data by RT−qPCR. The RNA-seq results of DEGs were used as controls.

**Figure 5 pathogens-12-01428-f005:**
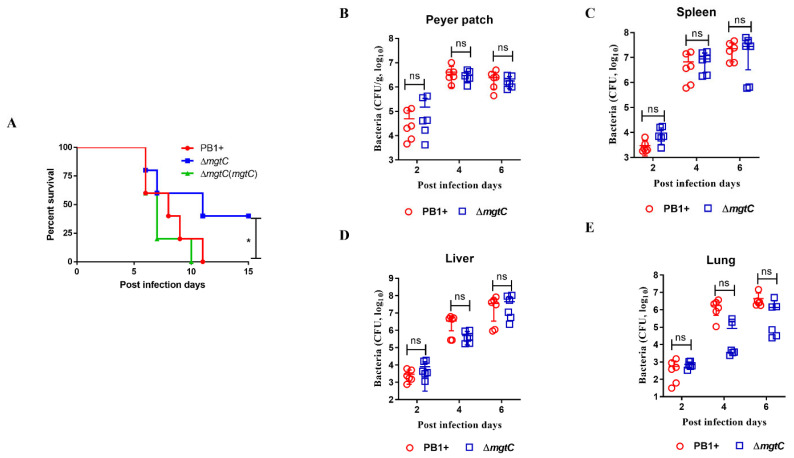
Virulence of the Δ*mgtC* mutant and kinetics of bacterial dissemination in mice. (**A**) Survival of Balb/c mice (n = 5 per group) that were gavaged with 5 × 10^8^ CFUs WT PB1+, Δ*mgtC*, and Δ*mgtC*(C-*mgtC*), respectively. The experiment was performed twice with similar results and pooled together. Mortality and morbidity were recorded in surviving mice for 15 days post-infection. The log-rank test was used for analysis of the survival curves. * *p* ≤ 0.05. (**B**–**E**) Kinetics of bacterial burden in mice administrated orally with 5 × 10^8^ CFU of WT PB1+, and Δ*mgtC* mutant, respectively. Bacterial titers in the (**B**) Peyer’s patches, (**C**) spleens, (**D**), livers and (**E**) lungs of infected mice at days 2, 4, and 6 post-infection. The experiment was performed twice with an equal numbers of animals (3 mice in each group); graphs represent pooled results. Statistical significance among groups were analyzed using a two-way ANOVA multivariant applying Tukey’s post hoc test. Mean value bars ± SD.

**Table 1 pathogens-12-01428-t001:** Strains and plasmids used in the present study.

Strains and Plasmids	Characteristics	Source of Reference
Strains		
*E. coli* strain		
DH5α	F^–^ φ80*lac*ZΔM15 Δ(*lac*ZYA-*arg*F)U169 *rec*A1 *end*A1 *hsd*R17(r_K_^−^, _K_^+^) *pho*A *sup*E44 λ^−^ *thi*-1 *gyr*A96 *rel*A1	Invitrogen
χ7213	*thi-1 thr-1 leuB6 fhuA21 lacY1 glnV44* Δ*asdA4 recA1* RP4 2-Tc::Mu [λ*pir*]; Km^r^	[24]
*Y. pseudotuberculosis* strain		
PB1+	Wild-type, serotype O:1b	Lab collection
PB1+ pYV^−^	Lack of virulence plasmid pYV	This study
YPtbS33	Δ*mgtC* in PB1+ strain	This study
YPtbS34	Δ*mgtC* in PB1+ pYV^−^ strain	This study
YPtbS35	Δ*YPTS_2313* in PB1+ strain	This study
YPtbS36	Δ*YPTS_3071* in PB1+ strain	This study
YPtbS37	Δ*YPTS_2313*Δ*YPTS_3071* in PB1+ strain	This study
YPtbS33(C-*mgtC*)	Amp^r^, Δ*mgtC* strain carrying the complementary plasmid pYA4454-*mgtC*	This study
YPtbS34 (C-*mgtC*)	Amp^r^, Δ*mgtC* pYV^−^ strain carrying the complementary plasmid pYA4454-*mgtC*	This study
YPtbS35 (C-*YPTS_2313*)	Amp^r^, Δ*YPTS_2313* strain carrying the complementary plasmid pYA4454-*YPTS_2313*	This study
YPtbS36 (C-*YPTS_2307*)	Amp^r^, Δ*YPTS_3071* strain carrying the complementary plasmid pYA4454-*YPTS_3071*	This study
Plasmids		
pRE112	Suicide vector, Cm^r^, mob^−^ (RP4)R6K *ori*, *sacB*	[25]
pYA4454	Complemental plasmid, Amp^r^	[26]
pSMV55	Cm^r^, pRE112 plasmid containing the Δ*mgtC* fragment	This study
pSMV56	Cm^r^, pRE112 plasmid containing the Δ*YPTS_2313* fragment	This study
pSMV57	Cm^r^, pRE112 plasmid containing the Δ*YPTS_3071* fragment	This study
pSMV58	Amp^r^, pYA4454 plasmid containing the *mgtC* gene	This study
pSMV59	Amp^r^, pYA4454 plasmid containing the *YPTS_2313* gene	This study
pSMV60	Amp^r^, pYA4454 plasmid containing the *YPTS_3071*gene	This study

Cm, chloramphenicol; Amp, ampicillin.

**Table 2 pathogens-12-01428-t002:** List of DEGs between PB1+ and the Δ*mgtC* mutant within macrophages.

Gene Number	Gene Name	Fold Change Δ*mgtC/*PB1+	*p*-Value
YPTS_0907	Transporter substrate-binding domain-containing protein	6.494781	1.04 × 10^−8^
YPTS_1642	Sugar ABC transporter substrate-binding protein	6.125791	1.59 × 10^−6^
YPTS_3843	Malate synthase A	5.255804	1.16 × 10^−10^
YPTS_2011	Aspartate aminotransferase family protein	5.083721	1.85 × 10^−7^
YPTS_3842	Isocitrate lyase	5.081315	4.39 × 10^−9^
YPTS_2313	Acid shock protein	−3.712933	8.92 × 10^−24^
YPTS_0225	Glycerol-3-phosphate dehydrogenase subunit GlpB	−3.838456	7.86 × 10^−35^
YPTS_0224	Anaerobic glycerol-3-phosphate dehydrogenase subunit A	−4.235415	9.02 × 10^−37^
YPTS_0091	Aquaporin family protein	−4.564323	1.58 × 10^−31^
YPTS_3071	Acid shock protein	−5.886112	1.59 × 10^−9^

DEGs: differentially expressed genes.

## Data Availability

No new data were created or analyzed in this study. Data sharing is not applicable to this article.

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
