# Peer review of "Impact of MgtC on the Fitness of Yersinia pseudotuberculosis"

_pathogens, 2023, doi:10.3390/pathogens12121428_

Round 1

Reviewer 1 Report

Comments and Suggestions for Authors

In this manuscript, the authors investigated the impact of MgtC mutation in Yersinia pseudotuberculosis using strain PB1+. It has been shown before that MgtC is important for intracellular bacterial pathogenesis, and that the functions of Yersinia MgtC closely relate to that of Salmonelle. The authors showed here that deletion of MgtC impact PB1+ growth in low Mg++ condition and intracellular survival. Importantly, in mouse infection model, they showed a defect of MgtC-deletion strain in causing death of the infected animal. The authors interpreted that the MgtC is important for virulence in the earlier stage of infection, however, their colonization results did not quite support this point. In summary, the manuscript represents some progress in the knowledges of MgtC in Yersinia pathogenesis. Specific points that may improve the manuscript are listed below:

1.     The phylogenetic analysis as shown in Fig.1 can be improved by providing details such as the software or method used. In addition, the gene or protein name should be listed, such as NP_462663.1 for Salmonella MgtC.

2.     Are there differences in the sequence or expression levels in MgtC among different strains of Y. pseudotuberculosis? In Discussion, the authors indicated that there are differences among the strains in intracellular survival in macrophages. Does MgtC play any role? How about the genes showed expression differences after MgtC-deletion?

3.     There is a clear difference in the survival of the animals, however, no significant differences were identified in the colonization levels at days 2, 4, 6 post infection. Are there differences in the recruitment of cells such as macrophages or neutrophils? Alternatively, for strains that are defective in intracellular survival, does MgtC impact virulence in animals?

Comments on the Quality of English Language

The language of the manuscript can be improved. There are numerous small points that can be polished to make the manuscript clearer and easier to read.  For example, in Abstract, line 9, “MgtC is known “as” a key subversion…” 

Author Response

Comments and Suggestions for Authors

In this manuscript, the authors investigated the impact of MgtC mutation in Yersinia pseudotuberculosis using strain PB1+. It has been shown before that MgtC is important for intracellular bacterial pathogenesis, and that the functions of Yersinia MgtC closely relate to that of Salmonelle. The authors showed here that deletion of MgtC impact PB1+ growth in low Mg++ condition and intracellular survival. Importantly, in mouse infection model, they showed a defect of MgtC-deletion strain in causing death of the infected animal. The authors interpreted that the MgtC is important for virulence in the earlier stage of infection, however, their colonization results did not quite support this point. In summary, the manuscript represents some progress in the knowledges of MgtC in Yersinia pathogenesis. Specific points that may improve the manuscript are listed below:

Response: Thank you very much for your affirmation and approval to this manuscript. Based on your excellent comments, the specific points were addressed and revised in the revised manuscript.

  1. The phylogenetic analysis as shown in Fig.1 can be improved by providing details such as the software or method used. In addition, the gene or protein name should be listed, such as NP_462663.1 for Salmonella MgtC.

Response: Thank you for your excellent suggestion. We re-constructed the phylogenetic tree (Fig.1) that listed the protein name and the NCBI Reference Sequence Number (showed at Line 199 in the revised manuscript).

Additionally, the software and method used in phylogenetic analysis (Fig.1) were respectively Molecular Evolutionary Genetics Analysis (MEGA) Software and Neighbor-Joining Tree (NJ) Method. Line 186 and Line 200 were listed the phylogenetic analysis software and method in the revised manuscript.

  1. Are there differences in the sequence or expression levels in MgtC among different strains of Y. pseudotuberculosis? In Discussion, the authors indicated that there are differences among the strains in intracellular survival in macrophages. Does MgtC play any role? How about the genes showed expression differences after MgtC-deletion?

Response: Thank you for your excellent comments. Firstly, we compared the MgtC/SapB family protein sequence by BLAST analysis in the different strains of Y. pseudotuberculosis, such as Yersinia pseudotuberculosis serotype IB (strain PB1/+) (studied in this manuscript)Yersinia pseudotuberculosis serotype O:1b (strain IP 31758)Yersinia pseudotuberculosis serotype I (strain IP32953) and Yersinia pseudotuberculosis serotype O:3 (strain YPIII), showing that there is no difference with 100% identity in the MgtC/SapB family protein sequence of these strains of Y. pseudotuberculosis. Secondly, in Discussion, we talked about the differences among the strains in intracellular survival in macrophages, and MgtC is not the critical factor for the differences, considering that the PB1ΔmgtC strain is still have the intracellular survival ability in macrophages (showed in Figure 3). Thirdly, we deleted the two genes showed expression differences after MgtC-deletion, YPTS_2313 and YPTS_3071, in Y. pseudotuberculosis PB1, and the intracellular survival ability of PB1ΔYPTS_2313 and PB1ΔYPTS_3071 in macrophages were similar to the wild-type strain PB1 (showed in Additional file 4), indicating that YPTS_2313 and YPTS_3071 also are not the critical factors for the differences. This is a very good point, and further studies are needed to uncover unknown mechanisms.

  1. There is a clear difference in the survival of the animals, however, no significant differences were identified in the colonization levels at days 2, 4, 6 post infection. Are there differences in the recruitment of cells such as macrophages or neutrophils? Alternatively, for strains that are defective in intracellular survival, does MgtC impact virulence in animals?

Response: Thank you for your excellent comments. Definitely, we also surprised that no significant differences were identified in the colonization levels. But the results in the Figure 5E showed that the lung tissues of several mice infected with PB1ΔmgtC strain displayed lower bacterial localization compared the wild-type PB1, which may be related to the recruitment of cells such as macrophages or neutrophils. In this study, we did not detect MgtC role of strains that are defective in intracellular survival for the virulence in animals, but this is a very good point. In the future, we will try to explore the unknown mechanisms of differences among the Y. pseudotuberculosis strains in intracellular survival in macrophages.

The language of the manuscript can be improved. There are numerous small points that can be polished to make the manuscript clearer and easier to read.  For example, in Abstract, line 9, “MgtC is known “as” a key subversion…” 

Response: Thank you for your excellent suggestion. We polished the manuscript. For example, Line 9 “MgtC is known as a key subversion factor.”  Line 11 “MgtC in Y. pseudotuberculosis might share the similar functions with it in other bacterial pathogens, such as Salmonella”.

Reviewer 2 Report

Comments and Suggestions for Authors

In this study, Li and Wang et al. have investigated the role of mgtC in the pathogenesis of Yersinia pseudotuberculosis. Previous publications have established that mgtC plays a role in the survival of numerous other bacterial pathogens in low Mg2+ conditions and inside macrophages, including Yersinia pestis (Ref 13) which is a close relative of Y. pseudotuberculosis. Results presented indicate that mgtC of Y. pseudotuberculosis is important for these phenotypes. Additionally, the authors present new results of RNAseq of the mgtC mutant in macrophages (Fig.4) and a mouse oral infection model (Fig.5).  The RNAseq results add new information to the field, and the mgtC mutant had a moderate but significant virulence defect in the mouse infection model. Together, the results extend understanding of mgtC function in Y. pseudotuberculosis. This manuscript is clearly written, with well described materials and methods, and represents the first description of a MgtC homolog in Y. pseudotuberculosis.

Minor:

Line 23: rephase sentence: "In-silico analysis, MgtC from Y. pseudotuberculosis might be transferred horizontally from its counterpart of Salmonella."

Line 215: I am curious that have you compared the expression of mgtC in the low Mg2+ and normal Mg2+ media?

The entire section should be carefully scrutinized for format correctness. Like Line 60 (37 °C) Line 95 (8000 x g), Line 116 (30 s)

In Fig3, the y axis is supposed be Log Bacteria CFU/well?

Comments on the Quality of English Language

.

Author Response

Comments and Suggestions for Authors

In this study, Li and Wang et al. have investigated the role of mgtC in the pathogenesis of Yersinia pseudotuberculosis. Previous publications have established that mgtC plays a role in the survival of numerous other bacterial pathogens in low Mg2+ conditions and inside macrophages, including Yersinia pestis (Ref 13) which is a close relative of Y. pseudotuberculosis. Results presented indicate that mgtC of Y. pseudotuberculosis is important for these phenotypes. Additionally, the authors present new results of RNAseq of the mgtC mutant in macrophages (Fig.4) and a mouse oral infection model (Fig.5).  The RNAseq results add new information to the field, and the mgtC mutant had a moderate but significant virulence defect in the mouse infection model. Together, the results extend understanding of mgtC function in Y. pseudotuberculosis. This manuscript is clearly written, with well described materials and methods, and represents the first description of a MgtC homolog in Y. pseudotuberculosis.

Response: Thank you very much for your affirmation and approval to this manuscript.

Minor:

Line 23: rephase sentence: "In-silico analysis, MgtC from Y. pseudotuberculosis might be transferred horizontally from its counterpart of Salmonella."

Response: Thank you for your excellent suggestion. Line11, we rephased sentence as “In-silico analysis, MgtC in Y. pseudotuberculosis might share the similar functions with it in other bacterial pathogens, such as Salmonella.”

Line 215: I am curious that have you compared the expression of mgtC in the low Mg2+ and normal Mg2+ media?

Response: Thank you for your excellent comments. Yes, we have compared the expression of mgtC in the low Mg2+ and normal Mg2+ media by qPCR analysis, the result showed similar results compared to the result in the high Mg2+ media.

The entire section should be carefully scrutinized for format correctness. Like Line 60 (37 °C) Line 95 (8000 x g), Line 116 (30 s)

Response: Thank you for your excellent comments. We scrutinized the format correctness. Such as Line 60 (37 °C), Line 95 (8000 × g), Line 116 (30 s).

In Fig3, the y axis is supposed be Log Bacteria CFU/well?

Response: Thank you for your excellent comments. We revised the y axis as Log Bacteria CFU/well in Fig3.

Round 2

Reviewer 1 Report

Comments and Suggestions for Authors

1.     The authors indicated that in Figure 5E, the lung tissues of several mice infected with PB1ΔmgtC strain displayed lower bacterial localization compared the wild-type PB1. By power analysis, to detect a difference equal to the value of 1 SD with a power of 90% and a significance of 5%, 11 mice are needed for each condition. The authors may want to include additional animals to clarify. 

2.     The authors may want to include the information in the manuscript that there is no difference in the published sequences of MgtC/SapB family protein from several strains of Y. pseudotuberculosis.

Comments on the Quality of English Language

no additional comments on this point. 

Author Response

Comments and Suggestions for Authors:

  1. The authors indicated that in Figure 5E, the lung tissues of several mice infected with PB1ΔmgtC strain displayed lower bacterial localization compared the wild-type PB1. By power analysis, to detect a difference equal to the value of 1 SD with a power of 90% and a significance of 5%, 11 mice are needed for each condition. The authors may want to include additional animals to clarify.

Response: Thank you for your excellent comments. In this study, the ΔmgtC mutant was slight attenuation in mice (40% mice were survival at 15 d.p.i.), and titers of the ΔmgtC mutant in a few mice that maybe will be survival at 15 d.p.i. were obviously lower than titers of WT PB1+ at day 2, 4 and 6 p.i., however, no significant differences were identified in the colonization levels at days 2, 4, 6 post infection, maybe caused by less mice number. I agree with your opinion about the power analysis. If more mice were used to analyze the localization titers of the ΔmgtC mutant, the significant difference will be showed up. Here we think this result is also reasonable, so we did not include additional animals to clarify. However, in the future study, we will pay attention for this point to the power analysis.

  1. The authors may want to include the information in the manuscript that there is no difference in the published sequences of MgtC/SapB family protein from several strains of Y. pseudotuberculosis.

Response: Thank you for your excellent suggestions. Based on your excellent suggestions, we included the information in the revised manuscript at Line 195. Described as “Additionally, there is no difference for the MgtC/SapB family protein sequence in the different strains of Y. pseudotuberculosis, such as Y. pseudotuberculosis serotype O:1b (strain PB1/+), Y. pseudotuberculosis serotype O:1b (strain IP 31758), Y. pseudotuberculosis serotype I (strain IP32953) and Y. pseudotuberculosis serotype O:3 (strain YPIII).”